# HSP90α Mediates Sorafenib Resistance in Human Hepatocellular Carcinoma by Necroptosis Inhibition under Hypoxia

**DOI:** 10.3390/cancers13020243

**Published:** 2021-01-11

**Authors:** Yan Liao, Yue Yang, Di Pan, Youxiang Ding, Heng Zhang, Yuting Ye, Jia Li, Li Zhao

**Affiliations:** 1School of Basic Medicine and Clinical Pharmacology, China Pharmaceutical University, Nanjing 211100, China; 1831090212@stu.cpu.edu.cn (Y.L.); 1721091065@stu.cpu.edu.cn (Y.Y.); pdpharm@gmc.edu.cn (D.P.); 1731090187@stu.cpu.edu.cn (Y.D.); 1821091168@stu.cpu.edu.cn (H.Z.); 2Pathology and PDX Efficacy Center, China Pharmaceutical University, Nanjing 211100, China; 1620194538@cpu.edu.cn (Y.Y.); 1620194559@cpu.edu.cn (J.L.)

**Keywords:** hepatocellular carcinoma, necroptosis, hypoxia, HSP90α, sorafenib resistance

## Abstract

**Simple Summary:**

Hypoxia is one of the characteristics of most solid tumors and induces cell resistant to chemotherapy. In this paper, we established a hypoxia model in both in vitro and in vivo to investigate the mechanisms of Sorafenib resistance in Hepatocellular carcinoma (HCC). Here, we observed that necroptosis could be an important target of Sorafenib in liver cancer and necroptosis blocking might be important in Sorafenib resistance under hypoxia. Mechanistically, our work suggests that HSP90α plays a pivotal role in Sorafenib-induced necroptosis by binding with necrosome. HSP90α could promote MLKL chaperone-mediated autophagy degradation in hypoxia, which subsequently decreased necroptosis. Consequently, the inhibition of necroptosis contributes to Sorafenib resistant. The Sorafenib resistance was reversed by HSP90α inhibitor-Demethoxygeldanamycin (17-AAG) in vivo and in vitro. This study highlights the important role of HSP90α in Sorafenib resistance under hypoxia microenvironment, and provides a potential therapy target for liver cancer.

**Abstract:**

As one of the most common malignancies worldwide, Hepatocellular carcinoma (HCC) has been treated by Sorafenib, which is the first approved target drug by FDA for advanced HCC. However, drug resistance is one of the obstacles to its application. As a typical characteristic of most solid tumors, hypoxia has become a key cause of resistance to chemotherapy and radiotherapy. It is important to elucidate the underlying mechanisms of Sorafenib resistance under hypoxia. In this study, the morphological changes of hepatocellular carcinoma cells were observed by Live Cell Imaging System and Transmission Electron Microscope; Sorafenib was found to induce necroptosis in liver cancer. Under hypoxia, the distribution of necroptosis related proteins was changed, which contributed to Sorafenib resistance. HSP90α binds with the necrosome complex and promotes chaperone-mediated autophagy (CMA) degradation, which leads necroptosis blocking and results in Sorafenib resistance. The patient-derived tumor xenograft (PDX) model has been established to investigate the potential therapeutic strategies to overcome Sorafenib resistance. 17-AAG inhibited HSP90α and presented obvious reversal effects of Sorafenib resistance in vivo and in vitro. All the results emphasized that HSP90α plays a critical role in Sorafenib resistance under hypoxia and 17-AAG combined with Sorafenib is a promising therapy for hepatocellular carcinoma.

## 1. Introduction

Hepatocellular carcinoma (HCC) is reported to be the fifth most common cancer, with the second highest mortality among all cancers in adults [1]. At present, the common treatment strategies for liver cancer are surgical resection, radiation therapies, and chemotherapy [2].

The multikinase inhibitor Sorafenib, originally developed as a Raf kinase inhibitor, targets not only the MAPK/ERK pathway, but also the vascular endothelial growth factor receptors (VEGF-R) and the platelet-derived growth factor receptor (PDGF-R) [3]. Sorafenib contributes to a survival benefit of patients through reducing tumor angiogenesis and increasing cancer cell apoptosis [4,5,6]. However, its efficacy has always been hampered by the occurrence of drug resistance [7,8,9], and HCC is much more difficult to cure after relapse. Therefore, the urgent problem to explore is the mechanism of Sorafenib resistance and to work out an effective treatment.

At present, it seems that necroptosis can be one of the important mechanisms of Sorafenib in the treatment of cancer [10,11,12,13]. Necroptosis is non-apoptotic cell death, which dependends on the receptor interacting protein kinase 3 (RIPK3). RIPK3 and RIPK1 can activate each other, promoting its conversion to an amyloid-like filamentous structure termed the necrosome, resulting in the recruitment of another necroptosis mediator, mixed lineage kinase domain-like (MLKL) [14,15]. Phosphorylated MLKL forms oligomers translocate to intracellular membranes and the plasma membrane, which eventually leads to membrane rupture. Recent evidence indicates that inhibition of caspase-dependent apoptosis sensitizes many cancer cells to necroptosis [11]. This has led to widespread interest in exploring necroptosis as an alternative strategy for anti-cancer therapy.

Hypoxia is one of the characteristics of most solid tumors, which plays an important role in the occurrence and development of cancers. Adaptation of tumor cells to hypoxia has important biological effects on drug resistance. Previous studies have reported that sustained Sorafenib treatment may promote hypoxia within tumors, which has been associated with Sorafenib resistance to HCC patients as well as subcutaneous mice model on HCC [16]. Hypoxia usually results in the resistance of various tumors to therapy through inducing the activation of the HIF signaling pathway and the survival of tumor cells [17,18]. Molecular chaperones are a heterogeneous class of proteins unified by their primary function of assisting the cellular proteome to achieve and maintain a conformationally mature and functional state [19,20], HSP90α is a chaperone protein that interacts with client proteins that it is closely related to cell apoptosis, metastasis, invasion and chemotherapy resistance, it protects cells from damage and stimuli, promotes tumor cells growth, make tumor cells tolerate chemotherapy, heat treatment and other traumatic stimuli, and finally leads to treatment failure. In HCC, HSP90α expression positively correlated with HIF-1. Down-regulation of HIF-1α or HIF-1β completely blocks HSP90α secretion, indicating that HIF-1 is a critical upstream regulator of HSP90α secretion [21]. In recent years, HSP90α also proved to play an important role in drug resistance under hypoxia [22,23,24,25]. Demethoxygeldanamycin (17-AAG) is a derivative of geldanamycin that is currently undergoing clinical development as a novel anticancer agent for the treatment of human cancers [26,27]. 17-AAG induces tumor apoptosis and inhibits tumor proliferation in leukemia cells and prostate cancer has already been studied. The combination of 17-AAG and oxaliplatin or capecitabine in colorectal cancer cell lines has been studied. 17-AAG in combination with paclitaxel on anaplastic thyroid carcinoma cells has also been reported [28,29,30]. In this study, we confirmed that necroptosis was one of the important reasons why Sorafenib attacks HCC. Besides, we elucidated that HSP90α binds with the RIPK1/RIPK3/MLKL complex to promote chaperone-mediated autophagy (CMA) degradation, which would be the main cause of Sorafenib resistance. 17-AAG, as a specific inhibitor of HSP90α, could overcome Sorafenib resistance on HCC. Combining 17-AAG with Sorafenib might be a potential therapeutic strategy to enhance Sorafenib efficacy for the treatment of HCC.

## 2. Results

### 2.1. Sorafenib Induced HCC Necroptosis

Soft agar assay (Figure 1A) and Flow cytometry (Figure 1B) were used to detect the inhibitory effect of Sorafenib on HCC cell lines. Results suggested that, after being treated with 10 μM Sorafenib for 24 h, HepG2 and Huh7 cells growth was decreased to 9.5% and 16%, compared to the control group, respectively. The dead cells were increased to 35.1% and 18.1%, respectively.

Figure 1C,D showed the morphological changes in HepG2 cells by Live Cell Imaging System and a Transmission Electron Microscope. We can see that cells experienced normal mitotic proliferation in the control group. While in the Sorafenib treated group, cells suffered a blocked proliferation, cellular content aggregates, and eventually swelled until dead (Figure 1C). More accurate views of various changes in cell death patterns were investigated by a Transmission Electron Microscope. As shown in Figure 1D, the healthy state of liver cancer cells was presented in the control group. Picture No. 1 showed the complete organelles like mitochondria and it has complete cell membrane structures. However, Sorafenib induced cell death, including necroptosis, apoptosis, and autophagic death, etc. In Picture No. 2, cell membrane and organelles that were destroyed are shown in the Sorafenib treated group. Cells swelled, cell content was released, and the morphology of the main cell main presented as necrosis. We also observed an apoptotic body in Picture No. 3 and autophagosomes in Picture No. 4. In Figure 1E,F, the expression of necroptosis marker proteins such as RIPK1, RIPK3, and MLKL were detected by western blot and immunofluorescence. The results showed that RIPK1, RIPK3, and MLKL were increased in treated cells, which strongly suggested that Sorafenib activated the necroptosis pathway. Additionally, the necroptosis marker proteins were detected in tumor tissues in the PDX model. WB (Figure 1G) and IHC (Figure 1H) results showed that Sorafenib induced necroptosis in vivo as well. Besides, the HepG2 cells xenograft model was used to confirm the results (Appendix A). All the results indicated that Sorafenib induced HCC necroptosis in vitro and in vivo.

Apoptotic cell death involves the engagement of pathways that result in the activation of caspase proteases that ultimately cause the morphological features of cell death. In contrast, necroptosis was recognized as a caspase-independent cell death that can be triggered by tumor necrosis factor (TNF) in the presence of a pan-caspase inhibitor such as zVAD-fluoromethylketone (VAD). To detect the necroptosis inducing effect of Sorafenib, Nec, a specific inhibitor of RIPK1 kinase was used. As shown in Figure 1I, Sorafenib induced obvious cell death even when VAD existed, and the effect was reversed by Nec, which suggested that the necroptosis inducing effect of Sorafenib on HCC cells was apoptosis independent. The morphological changes of HepG2 cells under Sorafenib were observed by Live Cell Imaging System (Figure 1J). Transmission Electron Microscope showed similar results (Appendix A). In the HepG2 xenograft model, Sorafenib inhibited tumor growth significantly, while Nec weakened the effect (Appendix A). All the results showed that Sorafenib induced hepatocellular carcinoma death partly via the necroptosis pathway in vivo and in vitro.

### 2.2. Hypoxia Contributed to HCC Resistance to Sorafenib

Hypoxia can promote proliferation, invasion, metastasis, apoptosis, drug resistance and other malignant biological behaviors of tumor cells. The IC50 values of Sorafenib under normoxia and hypoxia were tested (Figure 2A). In HepG2 cells, the IC50 was 12.8 μM under normoxia and 194.2 μM under hypoxia. In Huh7 cells, the values were 11.1 μM and 34.7 μM. Clone formation experiment also confirmed that hypoxia can induce Sorafenib resistant under hypoxia (Figure 2B). Further, the results of trypan blue staining also showed that cell death induced by Sorafenib was significantly reduced under the hypoxia condition (Appendix A). WB results proved that the necroptosis pathway induced by Sorafenib was attenuated under hypoxia (Figure 2C). The results of Co-IP also revealed that the capacity of RIPK1 binding with RIPK3/MLKL was upregulated in normoxia and decreased in the hypoxia microenvironment after Sorafenib treatment, which indicated that hypoxia reduced the activation necroptosis pathway of Sorafenib (Figure 2D).

HIF1α (hypoxia inducible factor-1) is a key regulator of many signals in tumor occurrence, development, and chemotherapy resistance. In order to figure out the effect of HIF1α on Sorafenib resistance, HIF1-α siRNA was used. Soft Agar Cloning experiment showed that HIF1α knockdown can enhance the inhibitory effect of Sorafenib (Figure 2E). Results in Figure 2F showed that RIPK3 and MLKL in HepG2 cells were upregulated by Sorafenib with HIF1α being silenced under hypoxia. In the PDX model (Figure 2G), HIF1α was much more expressed in internal hypoxia tumor tissue than in external normoxia tumor. Additionally, Sorafenib could not activate the necroptosis pathway in the tumor internal area, which means hypoxia blocked the necroptosis induced by Sorafenib in vivo.

### 2.3. Hypoxia Impeded the Distribution of RIPK1/RIPK3/MLKL Complex in Cytoplasm

The necroptosis key protein MLKL usually acts on lipid and cell membrane structures, causing membrane damage and eventually leading to cell death [31,32]. In order to figure out the mechanisms of Sorafenib resistance under hypoxia, the changes in MLKL expression and the complex of RIPK1/RIPK3/MLKL in cytoplasm were studied. In Figure 3A, under normoxia, RIPK1, RIPK3, and MLKL were increased by Sorafenib in cytoplasm. However, under hypoxia, necroptosis related proteins were down-regulated in cytoplasm. The RIPK1/RIPK3/MLKL complex was also detected by Co-IP (Figure 3B). Results showed that, in cytoplasma, Sorafenib increased the RIPK1/RIPK3/MLKL complex under normoxia and had no obvious effect under hypoxia. Therefore, Sorafenib could not increase necrosome in cytoplasm, which might be an important reason for Sorafenib resistance under hypoxia.

In this study, Sorafenib-induced necroptosis destroy the membrane structures in hepatocellular carcinoma cells. Sequently we detected the distribution of MLKL in several kinds of organelles intracellular which has abundant membranous structure, such as cell membrane, mitochondria, endoplasmic reticulum, lysosomes. As shown in Figure 3C–F, MLKL increased after Sorafenib treatment in normoxia and was located in various organelles. However, the location was decreased in organelles and increased in the nucleus under hypoxia. Similar results were observed in primary tumor cells (Appendix A). All the results indicated that MLKL was strongly located in the kinds of organelles that had membranous structures in the process of Sorafenib induced necroptosis; then, the damaged membranes finally lead to cell death. Hypoxia reduced MLKL expression in organelles, which might be one of the main reasons why necroptosis induced by Sorafenib was attenuated under hypoxia.

### 2.4. HSP90α Promotes Chaperone-Mediated Autophagy (CMA) Degradation by Directly Binding to MLKL in Hypoxia

To investigate whether hypoxia could affect the necroptosis protein degradation under Sorafenib treatment, protein synthesis inhibitor CHX was performed. As shown in Figure 4A, MLKL did not change when CHX was used in normoxia. While under hypoxia, MLKL was time-dependently decreased by Sorafenib. It seemed that MLKL was more susceptible to degrade under hypoxia than under nomoxia when cells were co-cultured with Sorafenib. Generally, there are three main ways that proteins are degraded: the proteasome pathway, caspase pathway, and autophagy-lysosome pathway. To verify which pathway contributed to the MLKL degradation under hypoxia, proteasome inhibitor MG132, caspase inhibitor z-VAD-fmk, and autophagy inhibitor chloroquine were used. As shown in Figure 4A and Appendix A, rather than MG132 or z-VAD-fmk, the chloroquine abrogated the degradation of MLKL, indicating that the autophagy-lysosome pathway was the main degradation of MLKL in the hypoxia microenvironment.

Chaperone-mediated autophagy (CMA) is a lysosomal pathway of proteolysis that is responsible for the degradation of 30% of cytosolic proteins [31,32]. HSP70 and HSP90 is the most essential component for protein transport across the lysosomal membrane in process of CMA [33,34]. They cannot only locate the substrate protein, but also recognize substrate protein. HSP90α can also be formed a polymer structure on the membrane of lysosome cavity side to maintain the stability of the transport complex. LAMP2/HSP70/HSP90α complex is the marker and finally identified as the lysosomal membrane receptor of CMA pathway to combine and transport substrate proteins. In Figure 4B–D, we observed lysosomes with lyso-tracker and detected the expression of HSP70, LAMP2, and HSP90α by immunofluorescence. Notably, compared normoxia, HSP70, LAMP2, and HSP90α, they were strikingly increased under hypoxia and were located in lysosomes. Similarly, WB results indicated that HSP70, LAMP2, and HSP90α were high expressed under hypoxia (Figure 4E). Furthermore, the capacity of HSP90α binding with RIPK3 and MLKL was much more in hypoxia than that in normoxia (Figure 4F). The HSP70/HSP90α/LAMP2/MLKL complex was also detected by Co-IP. We tested the capacity of MLKL binding with HSP70/HSP90α/LAMP2 being increased obviously under hypoxia conditions in HepG2 cells (Figure 4G). Similar results were found in primary tumor cells (Figure 4H). Besides, the capacity of LAMP2 binding with HSP70/HSP90α/MLKL was increased in hypoxia. LAMP2 and HSP90α were also highly expressed in the tumor internal hypoxia area compared to the normoxia tumor area in vivo; however, the expression of MLKL was decreased in hypoxia (Appendix A). Collectively, MLKL, as a customer, was recognized by the HSP70/HSP90α/LAMP2 transporter and was transported into lysosome, and then degraded in the end; all these results suggested that MLKL could be degraded under hypoxia through the chaperone-mediated autophagy (CMA) degradation pathway.

To test the role of HSP90α in the necroptosis pathway, siRNA was used. As shown in Figure 4I, there was a loss of HIF1α inhibited HSP90α, while a loss of HSP90α did not change the HIF1α expression. Besides, both inhibition of HIF1α and HSP90α can promote MLKL recovered under hypoxia. In Figure 4J, necroptosis related proteins were down-regulated obviously and HSP90α was highly expressed in primary resistant cells, which suggested that necroptosis was dull in Sorafenib-resistant tumor cells. Taken together, these results indicated that HSP90α played a direct and important role in Sorafenib resistance by blocking necroptosis.

### 2.5. 17-AAG Combining with Sorafenib Enhanced Necroptosis Pathway In Vitro

In order to study whether the inhibitory effect of HSP90α can improve the necroptosis induced by Sorafenib, 17-AAG, an inhibitor of HSP90α, was used. As shown in Figure 5A, when combined with 1 μM 17-AAG, the PI-positive cells were increased to 29.6% and 37.7%, respectively in HepG2 and Huh7. While in Sorafenib group, only about 10.73% and 19.6% of dead cells were increased in hypoxia. 17-AAG showed low effects on HCC cells when used alone.

MTT results in Figure 5B also showed that hepatoma cell lines were insensitive to Sorafenib in hypoxia; the survival of HepG2 and Huh7 was 41.08% and 36.4%, respectively, when 17-AAG was combined with Sorafenib. In general, the inhibitory effect of Sorafenib was increased significantly under hypoxia when 17-AAG was used. In Figure 5C, 17-AAG inhibited HSP90α and safeguard necroptosis induced by Sorafenib. Immunofluorescence presented a similar phenomenon in HCC cell lines (Figure 5D). All the results revealed that 17-AAG reversed the Sorafenib resistance in hypoxia and enhanced the inhibitory effect of Sorafenib on HCC.

### 2.6. HSP90α Could Be an Important Target in Sorafenib Resistance In Vivo

To further verify the pivotal role of HSP90α in Sorafenib Resistance in vivo, the HSP90α knockdown primary cell line was established. In the Sorafenib resistant xenograft model, we tested the effect of 17-AAG combined with Sorafenib and HSP90α knockdown combined with Sorafenib, the results indicated that Sorafenib reduced the tumor growth most significantly when HSP90α was inhibited (Figure 6A–C).

Compared to the control group, the tumor weight has also been significantly reduced, to 57% and 48% in the 17-AAG combined Sorafenib group and Sorafenib combined HSP90α knockdown group, respectively. While the Sorafenib alone has poor efficacy in resistant tumor. The expression of HIF1α/HSP90α and RIPK3/MLKL were also detected by immunofluorescence and IHC tissue sections (Figure 6D,E). Necroptosis related proteins were low expressed in resistant tumors and HIF1α/HSP90α was highly expressed. However, in ShHSP90α group, necroptosis induced by Sorafenib was surprisingly reactivated. All these results indicated that HSP90α can be a useful target of Sorafenib resistance therapy via recovering the necroptosis pathway in liver cancer.

### 2.7. Clinical Analysis of HIF1α/HSP90α as a Therapeutic Target

Results of Figure 7A showed that the overall survival of HSP90α, HIF-1α, RIPK1 and RIPK3 in Kaplan-Meier Plotter.

The database showed that the higher the HIF1α/HSP90α expression, the worse the patient prognosis. It looks like RIPK1 has a short survival when highly expressed and the expression of RIPK3 has no obvious effect on the survival of liver cancer. The HSP90α are the most significant of these genes, the p value is 0.0027. Additionally, HIF1α/HSP90α showed positive correlation in the GEPIA (Gene Expression Profiling Interactive Analysis) database (Figure 7B). The expression of HSP90α, HIF1α and RIPK1 was stronger in cancer tissues than in liver tissues; however, RIPK3 was expressed more in liver tissues (Figure 7C). RIPK1 may get close to cancer development. The role of RIPK3 in cancer development is still unclear. We also analyzed the samples of patients with liver cancer by IHC and WB in Figure 7D–F, and similar results were obtained. HIF1α/HSP90α and RIPK1 were almost highly expressed in liver cancer tissues. We found that, in most cases, the expression of RIPK3 in cancer tissues was lower than that in normal tissues, and there are still about 25% cases showing the opposite situation. In general, we believe that RIPK3 may have a negative regulatory effect on tumorigenesis, different from RIPK1, and it has been reported in the reference that RIPK1 is involved in the NF-κB signaling pathway to regulate cancer development. We think that the specific effects of the necroptotic signaling pathway on liver cancer requires more research. As a molecular chaperone, HSP90α plays important roles in proliferation, apoptosis, and the drug resistance of various biological behaviors. It is a regulator of P53, HIF1α, P23, HSF and so on, which can perform their functions in the development of cancer. HSP90α can not only regulate the HIF signaling pathway but also plays a crucial role in the necroptosis signaling pathway. As a systemic expression gene, HSP90α plays an important role in proliferation, apoptosis, and drug resistance of various biological behaviors, and it is a regulator of other proteins that can perform their functions. All these results suggested that HSP90α would be a promising and useful target in hepatocellular carcinoma therapy.

## 3. Discussion

At present, the effective treatment of liver cancer is still very limited and the recurrence rate of liver cancer is still high. Sorafenib remains the only FDA-approved systemic drug for patients with advanced HCC. In fact, both the direct inhibitory effect of Sorafenib on tumor cells and on angiogenesis are very important. Many articles summarized mechanisms of Sorafenib. Sorafenib suppresses tumor angiogenesis and proliferation, and induces tumor cell apoptosis. The emergence of Sorafenib resistance is a gradual process. After a long-term use of Sorafenib, ischemic and hypoxic will appeared in tumor area due to the inhibition of microvessels. The inhibition of Sorafenib on tumor cells is weakened, and then drug resistance is further developed. In our study, we found that MLKL was degraded under hypoxia. Necroptosis was blocked, which results in Sorafenib resistance. The understanding of the mechanism of Sorafenib resistance will enable us to use it more effectively in clinics.

Here, we demonstrated that necroptosis would be one of the main targets of Sorafenib on HCC. Necroptosis related proteins RIPK1, RIPK3 and MLKL were highly expressed in hepatocellular carcinoma cells and tumor tissues with Sorafenib treatment, which means that Sorafenib activated the necroptosis pathway in vitro and in vivo. Hypoxia is a hallmark of solid tumors due to the rapid growth of cancer cells and the abnormal angiogenesis. Recent studies have confirmed that the adaptation of tumors to hypoxic microenvironment is not only to maintain the survival or growth of tumors, but also play an important role in drug resistance. In the present study, we demonstrated that hypoxia rendered resistance to Sorafenib in human HCC cells by attenuating necroptosis. We observed that abundant HSP90α binds with necrosome directly, and MLKL was found degraded by autophagy lysosomal degradation pathway in hypoxia. Further study revealed that the expressions of HSP70/HSP90α/LAMP2/MLKL complex was increased under hypoxia and was located in lysosomes. Therefore, all our results presented that MLKL was degraded under hypoxia, which resulted in the decreasing of the RIPK1/RIPK3/MLKL complex. Then necroptosis was interrupted and drug resistance appeared. Clinical data and patient samples also suggest that HSP90α expression in hepatocellular carcinoma was associated with prognosis and Sorafenib resistance. The blockage of HSP90α can significantly overcome Sorafenib resistance under hypoxia in vitro and in vivo.

17-AAG, a derivative of geldanamycin, is currently undergoing clinical development as a novel anticancer agent for the treatment of human cancers. It has been reported that 17-AAG could be an effective anticancer drug, whether used alone or in combination with other drugs [35,36]. Here, we found that 17-AAG decreased HSP90α, and then safeguarded the activated necroptosis fluently. 17-AAG combining with Sorafenib showed great inhibitory effects in HCC in vivo and in vitro. In fact, 17-AAG combined with Sorafenib increased both apoptosis and necroptosis. Besides, many researchers have proved that 17-AAG could inhibit cell proliferation, tumor metastasis and inducing cell apoptosis in many cancers; in this study, we focus on 17-AAG combined with Sorafenib, thus enhancing the necroptosis of HCC. Overall, our study demonstrates that Sorafenib induced necroptosis is a considerable pathway in liver cancer and HSP90α plays a critical role in Sorafenib resistance under hypoxia by blocking necroptosis. 17-AAG combined with Sorafenib is a promising therapy for hepatocellular carcinoma.

## 4. Materials and Methods

### 4.1. Reagents

Sorafenib (Bay 43-9006, Sigma-Aldrich), 17-AAG (Tanespimycin, MCE, HSP90α inhibitor), Z-VAD-FMK (Selleck, Caspase Inhibitor), Necrostatin-1 (Selleck, RIPK1 inhibitor), Necrosulfonamide (Selleck, MLKL inhibitor).

MTT (3-(4,5-dimethylthiazol-2-yl)-2,5-diphenytetrazoliumbromide) was obtained from Fluka Chemical Corp (Ronkonkoma, NY, USA) and was dissolved in 0.01 M phosphate-buffered saline (PBS). Antibodies against RIPK1(A7414), RIPK3(A5431), MLKL(A5579), HIF-1α(A16873) were purchased from Abclonal Technology (Wuhan, China). Antibodies against, HSP90α(BS6461), were products of Bioworld Technology (USA). Antibodies against AFP (4448) and LAMP2 (49067) were products of Cell Signaling Technology (Beverly, MA, USA). Antibodies against Lamin A (sc-293162), β-Tubulin (sc-166729) and β-Actin (sc-8432) were products of Santa Cruz Biotechnology (USA). Normal mouse and rabbit IgG-HRB secondary antibodies were purchased from Santa Cruz Biotechnology.

### 4.2. Cell Culture

In this study, different human HCC cell lines (Huh7, HepG2) were purchased from the Shanghai Institute of Cell Biology, Chinese Academy of Sciences (Shanghai, China). The human HCC cells were cultured in Dulbecco’s Modified Eagle Medium (Gibco, Carlsbad, CA, USA) with 10% (*v*/*v*) heat-inactivated fetal bovine serum (Gibco, Paisley, Scotland), 100U/mL streptomycin and 100 U/mL penicillin at 37 °C with 5% CO_2_. Primary tumor cells were isolated from tumor tissues and were cultured with the special medium for primary human liver cancer culture medium (iCell Bioscience Inc, Shanghai) with 10% fetal bovine serum.

Cells were maintained at 37 °C in a humidified incubator containing 20% O_2_, 5% CO_2_ and 75% N_2_ in normoxia. The hypoxic condition was achieved at 37 °C with a gas mixture containing 1% O_2_, 94% N_2_ and 5% CO_2_ in a humidified atmosphere.

### 4.3. Clinical Samples

Fresh primary liver cancer tissue and adjacent nontumor liver tissue samples were obtained from HCC patients undergoing hepatectomy, hepatic tumor ablation or percutaneous transhepatic biopsy and similar liver surgery at Jiangsu Cancer Hospital and NanJing Drum Tower Hospital.

### 4.4. PDX (Patient-Derived Tumor Xenograft) Model and Sorafenib—Resistance Model In Vivo

PDX model was constructed by the transplantation of tumor tissue from patients into severe NSG immunodeficiency mice; 5-6-week-old NSG mice were purchased from Vital River Laboratory Animal Technology (Beijing, China). Liver cancer samples were inserted into the mice’s armpit, and were then fed with normal feed and drinking water. As shown in Appendix A, when the tumor size reached 400–500 mm^3^, tumor tissues were stripped and inoculated into other immunodeficiency mice, which was the first generation. For the inducing resistant group, when the tumor size reached 100mm^3^, the mice were given an oral dosage of Sorafenib 80 mg/kg once a day. At the fourth generation, the Sorafenib resistance was detected obviously. Data in Appendix A showed the successive processes of Sorafenib inducing resistance. The sensitive primary cells were extracted from the tumors of non-treated group. The resistant primary cells were extracted from the tumors of the Sorafenib induced resistant group. The drug resistance was tested by MTT assay (Appendix A), flowcytometry (Appendix A), and trypan blue staining (Appendix A). The morphology of primary resistant cells was also observed by Transmission Electron Microscope. Sorafenib could not induce cell death, including necroptosis in primary resistant cells (Appendix A).

### 4.5. Animal Studies

5–6-week-old female BALB/c nude mice were purchased from the SLAC Laboratory (Shanghai, China). The transplanted tumors were induced by subcutaneous injection into the flanks of the mice with 3.0 × 10^6^/0.1 mL HepG2, Huh7 cells or primary tumor cells. A couple of days later, the tumor volume was measured by micrometer calipers, according to the size of the tumor, and the mice were average divided into groups. Additionally, mice were treated with 60 mg/kg Sorafenib every 2 days orally. Nec (1.65 mg/kg) and 17-AAG (25 mg/kg) was administered intra-peritonelly twice a week. 3 weeks later, the mice were killed, and the tumor xenografts were removed and measured. Tumor volume (TV) was calculated using the following formula: TV (mm^3^) = D/2 × d^2^, where D is the longest diameter and d is the shortest diameters.

All experiment animals were raised in air-conditioned rooms under controlled lighting (12 h light/day) and were provided with food and water at our discretion. Animal care and surgery protocols were approved by the Animal Care Committee of China Pharmaceutical University. All the animals were treated and used in a scientifically valid and ethical manner.

### 4.6. Western Blot Analysis

Cells were washed with cold PBS and lysed in RIPA Lysis buffer (ThermoFisher, ShangHai, China) containing protease/phosphatase inhibitors. After the lysates concentration was determined by the BCA protein assay (Pierce, Rockford, IL, USA), an equal amount of denatured proteins were subjected to SDS-PAGE gel electrophoresis and then transferred onto a nitrocellulose membrane (PallCor, Arroyto, Cordoba, Argentina), which was blocked by 5% non-fat milk in PBS, followed by incubation at 4 °C with specific primary antibodies overnight. Then, membranes were incubated with HRP goat anti-rabbit immunoglobulin G (IgG; H + L) or anti-mouse IgG (H + L) secondary antibody (Biosharp) for 1 h and finally visualized with chemiluminescence (ThermoFisher). Immunoreactive proteins were detected with the Tanon-5200 Multi Automatic chemiluminescence/fluorescence image analysis system (Tanon-5200, Tanon). All uncropped images were shown in Appendix A.

### 4.7. Immunohistochemistry (IHC)

Paraffin embedding sections were heated at 60 °C for half an hour and were dewaxed. For the IHC assay, tissue sections were incubated with 0.3% Triton-X 100 for 20 min after antigen retrieval. Then, the solution containing goat serum was used to block nonspecific binding sites. Next, primary antibodies were added and incubated at 4 °C overnight. Before interacting with DAB solution, tissues were incubated with biotin-labeled secondary antibodies at room temperature for 30 min. Finally, tissues were stained with hematoxylin and covered by neutral gum—the process was performed with standard techniques. The immunohistochemistry kit was purchased from Shanghai Yeasen BioTechnologies co., Ltd.

### 4.8. Immunofluorescence Assay

The cells were seeded onto cover glasses in a 6-well plate and fixed by 4% PFA for 15 min, followed by permeabilization using 0.3% Triton-X 100 for 15 min. Then, the cover glasses were blocked with 3% bovine serum albumin for 1 h and incubated with primary antibody at 4 °C overnight. Then, the cells were incubated with Alexa Fluor-conjugated secondary antibodies (Invitrogen, Carlsbad, CA, USA) for 1 h and stained by DAPI for 20 min. The confocal microscope was employed to photograph the protein expressions or location in the cells. Various organelle fluorescent probes such as ER-Tracker (KeyGen Biotech, NanJing, China) and Mito-Red (KeyGen Biotech, NanJing, China) were incubated before paraformaldehyde was fixed and the next steps were the same as above. After 30 min incubation at 37 °C in the dark, coverslips were fixed in 4% paraformaldehyde for 30 min. The images were captured with the Olympus FV1000 confocal microscope.

### 4.9. Cell Transfection

HIF1-α siRNA was purchased from Santa Cruz Biotechnology. SiRNA transfections were performed according to the manufacturer’s instructions, using Lipofectamine 2000 reagent (Invitrogen). After that, the transfected system was removed, and it was not until 24 h cultured in normal media that the cells were used for further experiment.

To establish HSP90α-knockdown cells, 293T cells were transfected with lentivirus plasmid, together with psPAX2 and pMD2G packing plasmids by X-treme GENE 9 (Roche) for 12 h Fresh, with medium added to 293T cells to replace the previous medium. Then, the supernatants of 293T cells were collected and mixed with fresh medium to infect tumor primary cells, along with 8 μg/mL polybrene. On the next day, supernatants were replaced with fresh medium containing 2 μg/mL puromycin. 2 μg/mL of concentration was maintained for more than one week and the puromycin-resistant cells were isolated and used for further experiments.

### 4.10. Cytoplasmic and Nuclear Protein Extraction 

The cells were normally collected and the cells were added with buffer A and mixed evenly. After that, the cells were cracked on ice for 15 min, and were mixed with intermittent shocks. At 4 ℃, 13,000 rmp, 5 min, the transferred supernatant was stored as the cytoplasmic part, washed three times with buffer A, and after washing the residual, the transferred supernatant was lysed with buffer B for 10 min. At 4 ℃, 13,000 rmp, 10 min, the transferred supernatant was the nuclear component; the protein concentration was measured by BCA, and after protein denaturation, it was stored with −20 ℃ or subsequent WB experiments.

### 4.11. Co-Immunoprecipitation (Co-IP)

The nuclear extracts were incubated with 1 μg control anti-IgG and 20 μL of Protein A/G PLUS-Agarose (Santa Cruz) at 4 °C for 30 min. After eliminating beads by centrifugation at 2500 rpm, 10 μL primary antibody was incubated with nuclear extracts at 4 °C for 1 h and 20 μL beads were added to rotate at 4 °C overnight. On the next day, samples were centrifugated at 2500 rpm for 5 min at 4 °C, and then the supernatants were carefully discarded. PBS was used to wash the pellets three times, and the samples were mixed with 20 μL 2 × loading buffer and boiled for 8 min. Finally, samples were analyzed by SDS-PAGE.

### 4.12. Flowcytometry Analysis

Cells were harvested and stained with the Annexin V/PI Cell Apoptosis Detection Kit (Vazyme Biotech Co.,Ltd, NanJing, China) according to the manufacturer’s instructions. Data were analyzed by FlowJo version 10. All the bar charts were analyzed by GraphPad Prism 6 software (GraphPad Software, San Diego, CA, USA).

### 4.13. Statistical Analyses

The data shown in the study were expressed as means ± standard errors (SEM) from at least three independent experiments, each in triplicate samples for individual treatments or dosages. Statistical analyses were performed using ANOVA, coupled with a post hoc test.

## 5. Conclusions

Overall, our findings identify that Sorafenib induced necroptosis in liver cancer and HSP90α plays a critical role in Sorafenib resistance under hypoxia by blocking necroptosis. Additionally, 17-AAG, combined with Sorafenib, is a promising therapy for hepatocellular carcinoma.

## Figures and Tables

**Figure 1 cancers-13-00243-f001:**
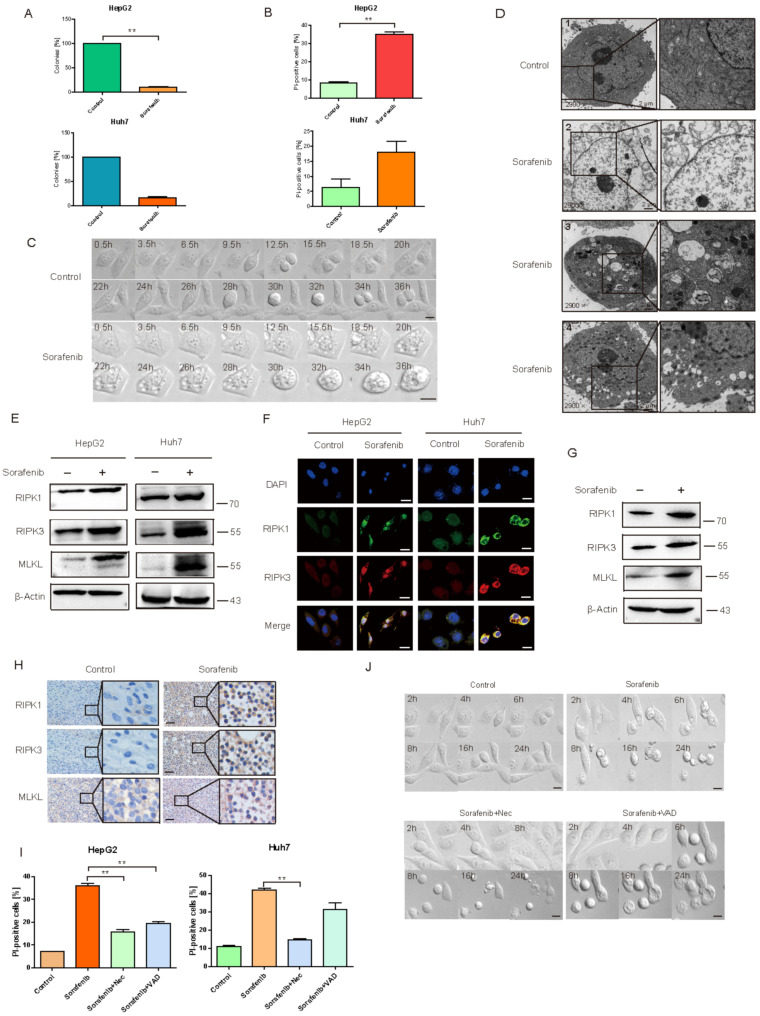
Sorafenib Induced HCC Necroptosis. (**A**) The inhibitory effect of Sorafenib on Hepatocellular carcinoma cell lines by Soft Agar Assay. The results were shown as mean ± SEM from three independent experiments. ** *p* < 0.01. (**B**) Detection of PI-positive cells in Huh7 and HepG2 cells with Sorafenib treatment by Flow Cytometry. The results were shown as mean ± SEM from three independent experiments. ** *p* < 0.01. (**C**) To observed the morphological changes of HepG2 cells by Live Cell Imaging System. Scale bar = 40 μm. (**D**) The different types of death in HepG2 cells induced by Sorafenib observed by Transmission Electron Microscope. (**E**,**F**) The expressions of necroptosis pathway in hepatocellular carcinoma cell lines after Sorafenib treated detected by Western blot and Immunofluorescence. Scale bar = 40 μm. (**G**,**H**) The expression of RIPK1, RIPK3 and MLKL were dectected in PDX-xenograft respectively by Western blot and IHC. Scale bar = 800 μm. (**I**) Detection of PI positive cells by flow cytometer to observe the dead cell when Nec and VAD exists. The results were shown as mean ± SEM from three independent experiments. ** *p* < 0.01. (**J**) Live cell Imaging System observed morphological changes in HepG2 cells when Nec and VAD exist. Scale bar = 40 μm.

**Figure 2 cancers-13-00243-f002:**
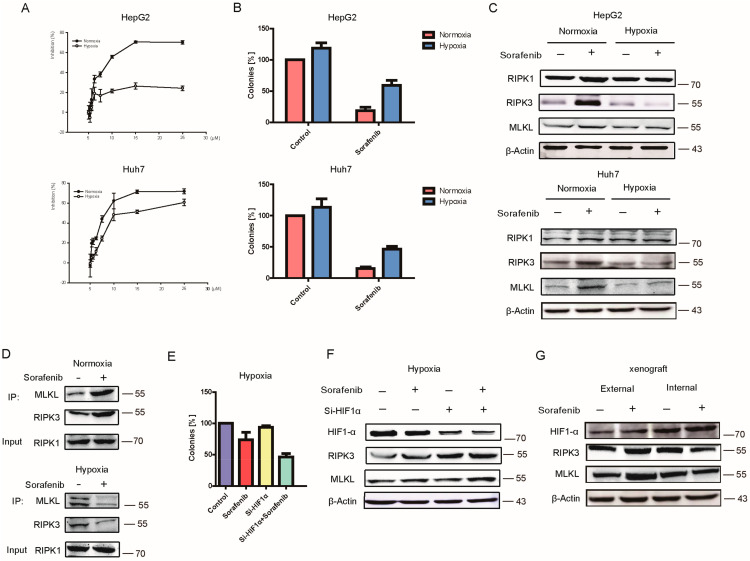
Hypoxia Induced Hepatocellular Carcinoma Resistant to Sorafenib. **(A)** Inhibition of Sorafenib under normoxia and hypoxia assessed by MTT assay. (**B**) Inhibition of Sorafenib under normoxia and hypoxia assessed by Clone formation experiment. (**C**) The expression of necroptosis proteins in normoxia and hypoxia detected by Western Blot. (**D**) The binding capacity of the necroptosis complex under normoxia and hypoxia. (**E**,**F**) HIF1α effect on Sorafenib observed by Soft Agar Cloning experiment. (**F**) HIF1α effect on Sorafenib in necroptosis pathway detected WB. (**G**) Detected HIF1α and necroptosis protein in PDX-xenograft tumor tissues by WB.

**Figure 3 cancers-13-00243-f003:**
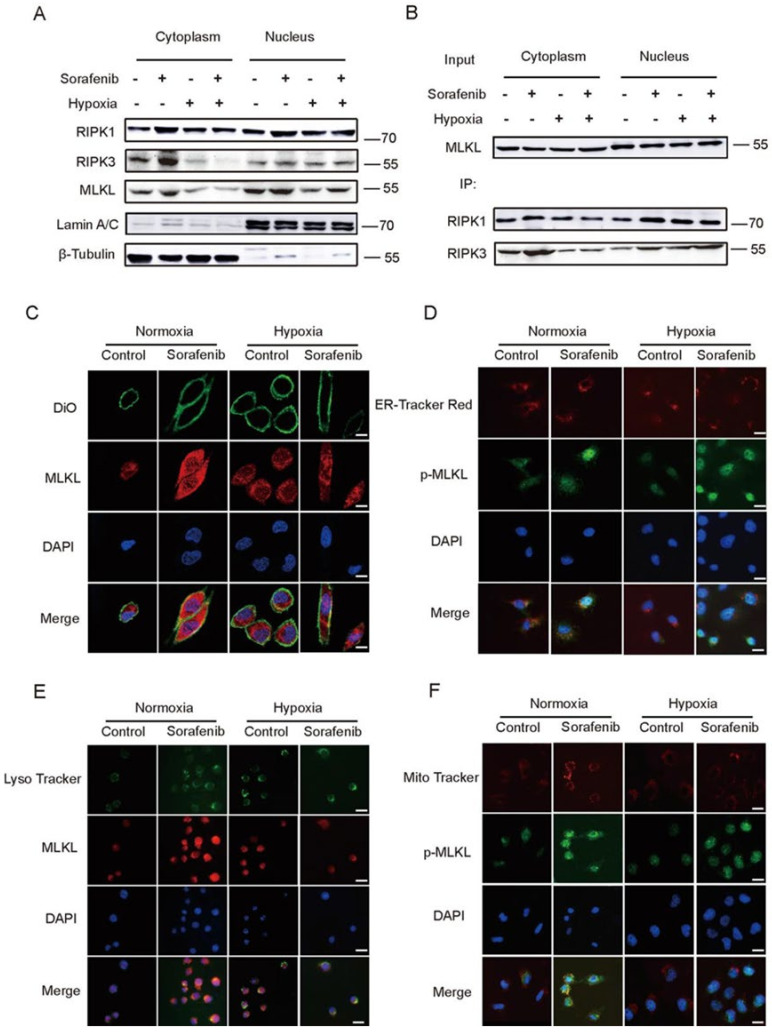
Hypoxia Impeded the Distribution of RIPK1/RIPK3/MLKL Complex in Cytoplasm. (**A**) Distribution of necroptosis proteins in cytoplasmic and nuclear observed by WB. (**B**) Distribution of RIPK1/RIPK3/MLKL complex in cytoplasm by Co-IP. (**C**–**F**) Expression of MLKL on different organelles, which has a rich membrane structure such as a cell membrane, mitochondria, endoplasmic reticulum, and lysosome observed by immunofluorescence. (**C**): Scale bar = 80 μm. (**D**–**F**): Scale bar = 40 μm.

**Figure 4 cancers-13-00243-f004:**
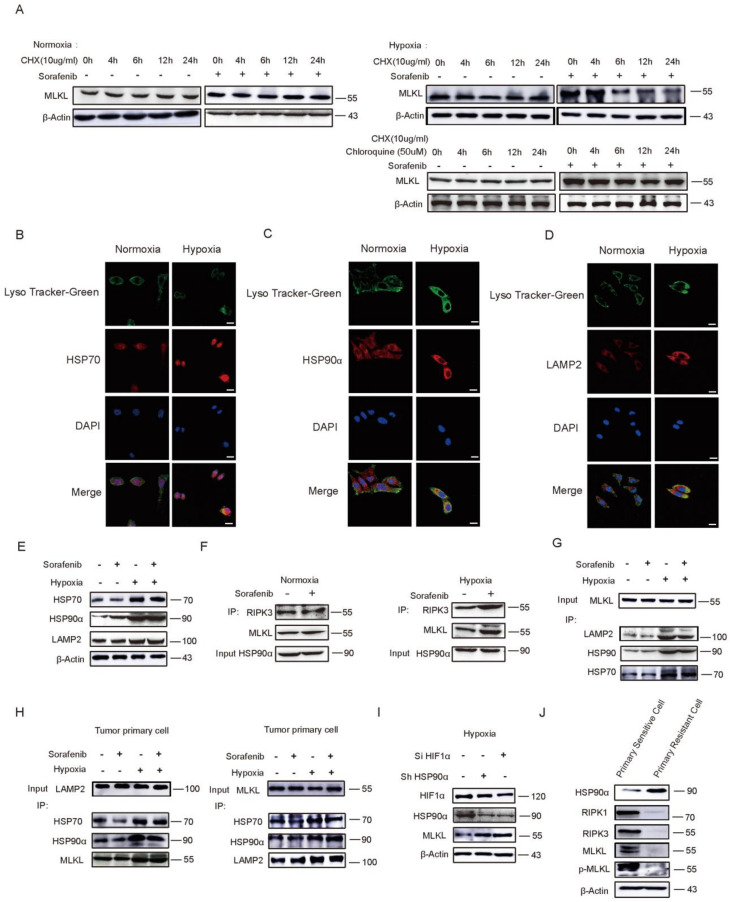
HSP90α Promotes Chaperone-Mediated Autophagy (CMA) Degradation by directly binding to MLKL in hypoxia. (**A**) The degradation stability of MLKL detected by Western blot. (**B**–**D**) The expressions of LAMP2, HSP90α and HSP70 located in lysosomal observed by immunofluorescence. Scale bar = 40 μm. (**E**) The expression of LAMP2, HSP90α and HSP70 in hypoxia. (**F**) HSP90α binding with necroptosis protein. (**G**,**H**) The expression of LAMP2/HSP90α/HSP70/MLKL complex detected by Western blot in HepG2 and primary tumor cells. (**I**) Silenced HIF-1α and HSP90α affected necroptosis in hypoxia. (**J**) Necroptosis pathway in primary resistant cells detected by Western blot.

**Figure 5 cancers-13-00243-f005:**
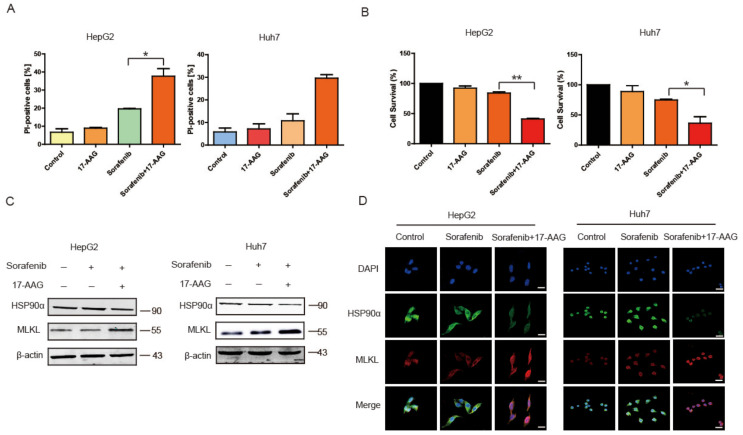
17-AAG Combined with Sorafenib Treatment Enhanced Necroptosis Pathway in vitro. (**A**) Detection of death cells under hypoxia by Flow Cytometry. The results were shown as mean ± SEM from three independent experiments. * *p* < 0.05, ** *p* < 0.01. (**B**) The survival of liver cancer cells under hypoxia assessed by the MTT assay. The results were shown as mean ± SEM from three independent experiments. * *p* < 0.05, ** *p* < 0.01. (**C**,**D**) The detection of HSP90α and MLKL by Western blot and Immunofluorescence. Scale bar = 40 μm.

**Figure 6 cancers-13-00243-f006:**
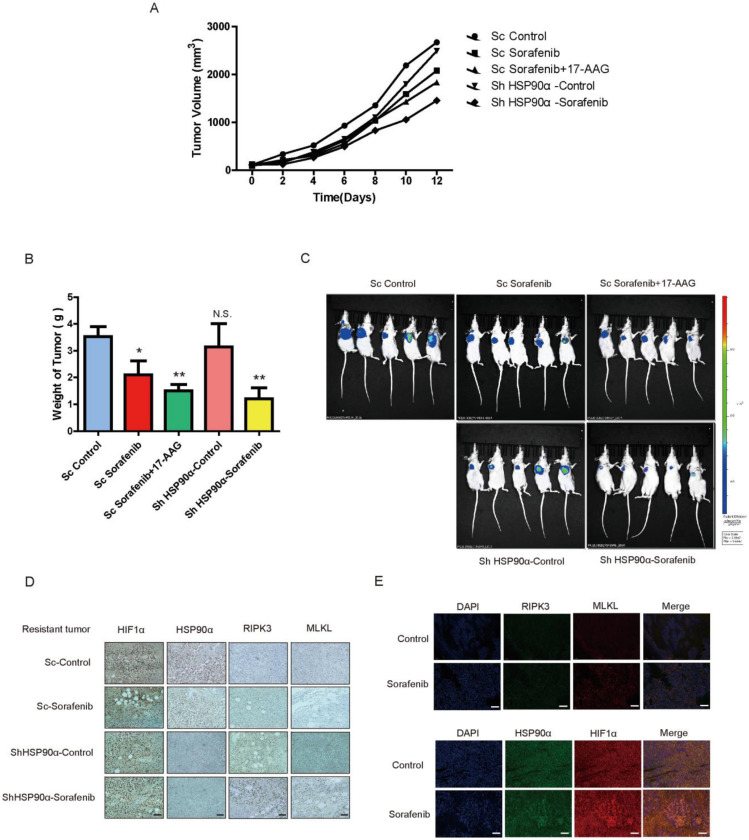
HSP90α can be a Useful Target in Sorafenib Resistance Therapy In Vivo. (**A**) Tumor volume was recorded in primary Sorafenib resistant xenograft model. (**B**) Tumor weight was recorded in primary Sorafenib resistant xenograft model. The results were shown as mean ± SEM from three independent experiments. * *p* < 0.05, ** *p* < 0.01, ns, not significant. (**C**) Bioluminescence images of the primary Sorafenib resistant xenograft model. (**D**,**E**) The expression of HIF1α, HSP90α and necroptosis pathway in tumors detected by IF and IHC. Scale bar = 800 μm.

**Figure 7 cancers-13-00243-f007:**
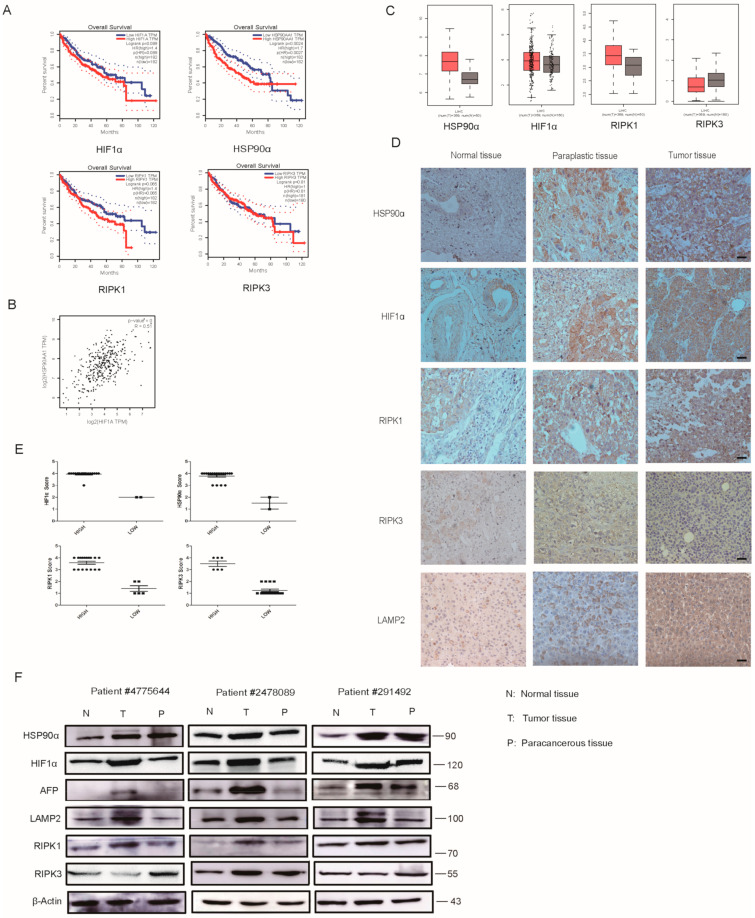
Clinical Analysis of HIF1α/HSP90α as a Therapeutic Target. (**A**,**B**) The overall survival and correlation of several proteins in GEPIA (Gene Expression Profiling Interactive Analysis) database. (**C**) Expressions of HSP90α, HIF-1α, RIPK1 and RIPK3 in GEPIA (Gene Expression Profiling Interactive Analysis) database. (**D**) Protein expressions in tumor, para-carcinoma, and normal tissues of patient samples detected by IHC. Scale bar = 500 μm. (**E**) Analysis the expressions of HSP90α, HIF-1α, RIPK1 and RIPK3 in patient samples. (**F**) Protein expressions in tumor, para-carcinoma, and the normal tissues of patient samples detected by WB.

## Data Availability

All data generated or analysed during this study are included in this article. The data presented in this study are available on request from the corresponding author.

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
