# Peer review of "HSP90α Mediates Sorafenib Resistance in Human Hepatocellular Carcinoma by Necroptosis Inhibition under Hypoxia"

_cancers, 2021, doi:10.3390/cancers13020243_

Round 1
Reviewer 1 Report
The manuscript entitled “HSP90α mediates Sorafenib resistance in human hepatocellular carcinoma by necroptosis inhibition under hypoxia” (Manuscript ID: cancers-1031282) reports the mechanism of HSP90α mediated Sorafenib resistance in human hepatocellular carcinoma by necroptosis inhibition under hypoxia. The experiments were thoughtfully designed and the results from both in vitro and in vivo studies support the conclusion of the manuscript. However, the manuscript requires careful editing before publication. Some suggested changes are given below.
- In the Materials and Methods section, line 318 of page 17, 17-AAG is a HSP90α inhibitor, not an HSP90 inhibitor. This could cause serious misunderstanding and confusions among readers of the journal.
- Line 165 of page 6. Authors showed that MLKL played its role in the cytoplasm and cell membrane, while the expression of p-MLKL induced by Sorafenib in the nucleus was still quite high under normoxia in Fig 3D. Please explain your observation in more detail.
- Line 241 of page 10. In Fig.5A, the combination of 17-AAG and Sorafenib increased the necroptosis of liver cancer cells under hypoxia. Is the synergistic effect of 17-AAG and Sorafenib only necroptosis-inducing?Are there some other mechanisms? The authors should address these questions in the manuscript.
- Line 275 of page 12. The conclusion of Fig.7F is “the expression of HSP90α and HIF1α was stronger in cancer tissues than in liver tissues, and RIPK3 was the opposite”. While in patient #2478089, RIPK3 expression in tumor is higher than in normal tissue and paracancerous tissue. Authors should explain this observation in the manuscript and modify their conclusion as necessary.
- The supplementary data needs to be re-organized. No one can understand what the individual data set is intended for. The materials downloaded from the manuscript site are composed of a folder name in Chinese and sub-sub-folders shown individual Western Blot results without any information. The individual links to supplementary data from Externally hosted supplementary files contain the same three figures. Why you need to have three links for the same information?
Last, but not least, there are lots of language mistakes/typos in the manuscript. For example, in Simple Summary: page 1, line 18 “…which sequently decreased necroptosis” is difficult to understand. Supplement data should be supplemental or supplementary data. The authors should seek help from a native English speaker to carefully edit the manuscript to ensure accurate communication of their work.
Author Response
Response to Reviewer 1 Comments
Point 1: In the Materials and Methods section, line 318 of page 17, 17-AAG is a HSP90α inhibitor, not an HSP90 inhibitor. This could cause serious misunderstanding and confusions among readers of the journal.
Response 1: Thank you for your kind reminder. The word "HSP90" has been replaced by"HSP90α" in the line 351.
Point 2: Line 165 of page 6. Authors showed that MLKL played its role in the cytoplasm and cell membrane, while the expression of p-MLKL induced by Sorafenib in the nucleus was still quite high under normoxia in Fig 3D. Please explain your observation in more detail.
Response 2: Thanks for this question. In Figure 3D, we observed the expressions of p-MLKL in HepG2 cells treated with Sorafenib. When necroptosis pathway was activated by Sorafenib under normoxia, the expressions of p-MLKL was increased in intracellular. p-MLKL was expressed both in cytoplasm and nucleus. In Figure 3D, we investigated the co-localization of p-MLKL and the endoplasmic reticulum. Compared with normoxia, Sorafenib-induced p-MLKL was reduced in hypoxia. It can be seen from the immunofluorescence image that there were still a large number of yellow light spots in the merge image, which proved that p-MLKL was located in the endoplasm after Sorafenib treated and necroptosis takes place in the endoplasmic reticulum under normoxia.
Point 3: Line 241 of page 10. In Fig.5A, the combination of 17-AAG and Sorafenib increased the necroptosis of liver cancer cells under hypoxia. Is the synergistic effect of 17-AAG and Sorafenib only necroptosis-inducing? Are there some other mechanisms? The authors should address these questions in the manuscript.
Response 3: Thank you for your comments. As a matter of fact, we observed the mortality of HCC after 17-AAG combined with Sorafenib by flow cytometry. Both apoptosis and necroptosis were increased in 17-AAG combining with Sorafenib as follows.
Besides, many researchers have proved that 17-AAG could inhibit the cell proliferation, tumor metastasis and inducing cell apoptosis in many cancers. And we have added related description in the discussion about 17-AAG in line 342-345.
Point 4: Line 275 of page 12. The conclusion of Fig.7F is "the expression of HSP90α and HIF1α was stronger in cancer tissues than in liver tissues, and RIPK3 was the opposite". While in patient #2478089, RIPK3 expression in tumor is higher than in normal tissue and paracancerous tissue. Authors should explain this observation in the manuscript and modify their conclusion as necessary.
Response 4: Thanks for your comments. The role of RIPK3 in cancer development is still unclear at present. The expression of RIPK3 in patient samples that we collected was shown in Fig. 7E. We found that in most cases, the expression of RIPK3 in cancer tissues was lower than that in normal tissues, there are still about 25% cases showing the opposite situation. In Fig. 7C , RIPK3 expressed more in normal tissues than in cancer tissues in GEPIA datebase, which is similiar with our results. And we have rephrased related results and conclusion in the line 291-297.
Point 5: The supplementary data needs to be re-organized. No one can understand what the individual data set is intended for. The materials downloaded from the manuscript site are composed of a folder name in Chinese and sub-sub-folders shown individual Western Blot results without any information. The individual links to supplementary data from Externally hosted supplementary files contain the same three figures. Why you need to have three links for the same information?
Response 5: Thank you for your advice. We have uploaded supplementary data and renamed the documents for replacement. The supplementary materials includs supplementary figures, WB original data, and ethical code.
Thanks again for all your comments and valuable suggestions of our manuscript.

Reviewer 2 Report
Sorafenib, a multikinase inhibitor , constitutes the only effective first-line drug approved for the treatment of advanced hepatocellular carcinoma (HCC). Despite its capacity to increase survival in HCC patients, its success is quite low in the long term owing to the development of resistant cells through several mechanisms. Among these mechanisms, the antiangiogenic effects of sustained sorafenib treatment induce a reduction of microvessel density, promoting intratumoral hypoxia and hypoxia-inducible factors (HIFs)-mediated cellular responses that favor the selection of resistant cells adapted to the hypoxic microenvironment. Clinical data have demonstrated that overexpressed HIF-1α and HIF-2α in HCC patients are reliable markers of a poor prognosis.
In this sense the novelty of this research is very low: since relevant papers on this subjects:
Zhu YJ, Zheng B, Wang HY, Chen L. New knowledge of the mechanisms of sorafenib resistance in liver cancer. Acta Pharmacol Sin. 2017 May;38(5):614-622. doi: 10.1038/aps.2017.5.
Méndez-Blanco C, Fondevila F, García-Palomo A, González-Gallego J, Mauriz JL. Sorafenib resistance in hepatocarcinoma: role of hypoxia-inducible factors. Exp Mol Med. 2018 Oct 12;50(10):1-9. doi: 10.1038/s12276-018-0159-1
Qiu Y, Shan W, Yang Y, Jin M, Dai Y, Yang H, Jiao R, Xia Y, Liu Q, Ju L, Huang G, Zhang J, Yang L, Li L, Li Y. Reversal of sorafenib resistance in hepatocellular carcinoma: epigenetically regulated disruption of 14-3-3η/hypoxia-inducible factor-1α. Cell Death Discov. 2019 Jul 19;5:120. doi: 10.1038/s41420-019-0200-8.
Niu L, Liu L, Yang S, Ren J, Lai PBS, Chen GG. New insights into sorafenib resistance in hepatocellular carcinoma: Responsible mechanisms and promising strategies. Biochim Biophys Acta Rev Cancer. 2017 Dec;1868(2):564-570. doi: 10.1016/j.bbcan.2017.10.002.
Jiang W, Li G, Li W, Wang P, Xiu P, Jiang X, Liu B, Sun X, Jiang H. Sodium orthovanadate overcomes sorafenib resistance of hepatocellular carcinoma cells by inhibiting Na+/K+-ATPase activity and hypoxia-inducible pathways. Sci Rep. 2018 Jun 26;8(1):9706. doi: 10.1038/s41598-018-28010-y.
Wu FQ, Fang T, Yu LX, Lv GS, Lv HW, Liang D, Li T, Wang CZ, Tan YX, Ding J, Chen Y, Tang L, Guo LN, Tang SH, Yang W, Wang HY. ADRB2 signaling promotes HCC progression and sorafenib resistance by inhibiting autophagic degradation of HIF1α. J Hepatol. 2016 Aug;65(2):314-24. doi: 10.1016/j.jhep.2016.04.019.
In contrast dose-dependence effects were not adressed in vivo experiments nor in vitro analyses.
17-AAG, a derivative of geldanamycin shoudl be tested in all measurement also alone without sorafenib in order to shown any effect on its own.
The role of protein expression and phosphorylation of HSP90α which may be linked to Sorafenib resistance in human hepatocellular carcinoma
Author Response
Response to Reviewer 2 Comments
Point 1: Sorafenib, a multikinase inhibitor, constitutes the only effective first-line drug approved for the treatment of advanced hepatocellular carcinoma (HCC). Despite its capacity to increase survival in HCC patients, its success is quite low in the long term owing to the development of resistant cells through several mechanisms. Among these mechanisms, the antiangiogenic effects of sustained sorafenib treatment induce a reduction of microvessel density, promoting intratumoral hypoxia and hypoxia-inducible factors (HIFs)-mediated cellular responses that favor the selection of resistant cells adapted to the hypoxic microenvironment. Clinical data have demonstrated that overexpressed HIF-1α and HIF-2α in HCC patients are reliable markers of a poor prognosis. In this sense the novelty of this research is very low: since relevant papers on this subjects.
Response 1: Thank you for your comments. HIFs regulate the occurrence and development of tumors, and it is known as a reliable marker of prognosis. HIFs have an important impact on resistance to chemotherapy. HSP90α also overexpressed under hypoxia. The role of HSP90α in Sorafenib resistance is still unclear. The aim of this study is to discover the mechnisms of HSP90α in Sorafenib resistance under hypoxia microenvironment. HSP90α binds with necrosome directly, and MLKL was found degraded by autophagy lysosomal degradation pathway under hypoxia eventually resulting in Sorafenib resistance. And we suggested 17-AAG, a HSP90α inhibitor, might be a potential therapeutic agent to enhance Sorafenib efficacy for the treatment of HCC.
Point 2: In contrast dose-dependence effects were not adressed in vivo experiments nor in vitro analyses.
Response 2: Thank you for your comments. MMT assay was used to detect the dose-dependence effects of Sorafenib. In futher in vitro experiments, a concentration of 50% ×IC50 was used. In the test in vivo, the doses of Sorafenib and 17AAG were selected by preliminary experiments. 80 mg/kg Sorafenib is effective dose in xenograft model. 25mg/kg 17AAG is a low toxicity dose.
Point 3: 17-AAG, a derivative of geldanamycin should be tested in all measurement also alone without sorafenib in order to shown any effect on its own.
Response 3: Thanks for your comments. In effeacy of 17AAG was reported in several articles. In this study, we have investigated the inhibitory effect of 17-AAG combined with Sorafenib on liver cancer. A low toxicity dose of 17-AAG was selected by our pre-experiment. We used 1 μM 17-AAG in vitro and 25mg/kg in vivo. There was slight inhibitor effects on HCC with 17-AAG alone. We found similar results in the following reference.
Calero, R.; Morchon, E.; Martinez-Argudo, I.; Serrano, R. Synergistic anti-tumor effect of 17aag with the pi3k/mtor inhibitor nvp-bez235 on human melanoma. Cancer letters 2017, 406, 1-11.
Liu, M.; Li, M.Y.; Zhou, Y.; Zhou, Q.; Jiang, Y.G. Hsp90 inhibitor 17aag attenuates sevoflurane-induced neurotoxicity in rats and human neuroglioma cells via induction of hsp70. J Transl Med 2020, 18.
Point 4: The role of protein expression and phosphorylation of HSP90α which may be linked to Sorafenib resistance in human hepatocellular carcinoma
Response 4: Thank you for your constructive comments. 17-AAG was designed for the ATP/ADP binding site of the N-terminal domain of HSP90α to inhibit the activity of HSP90α. In our study, we used 17-AAG to reverse Sorafenib resistance. ShRNA were also applied to decrease HSP90α expression. In Fig. 6A-C, both reduced and devitalized HSP90α improved effect of Sorafenib in vivo. In the group of shHSP90α conbining with Sorafenib, the inhibitory effect is the best. The reason might attribute to that shHSP90α reduced the protein expression, at the same time, phosphorylation of HSP90α was also decreased. All the results proved both protein expression and phosphorylation of HSP90α were linked to Sorafenib resistance in human hepatocellular carcinoma.
Thanks again for all your comments and valuable suggestions of our manuscript.

Reviewer 3 Report
The ms. by Liao et al. presents a wide array of data concerning the role of necroptosis and Hsp90 in the effects of Sorafenib on HCC cells, demonstrating the protective role of the stress protein on cancer and, hence, suggesting a potential role of Hsp90 inhibitors, combined with sorafenib, in HCC therapy.
Experiments are performed both in vitro and in vivo with a variety of techniques and approaches. Results are interesting and conclusions of potential translational value. However, I have some critical remarks.
a) Sorafenib is a multikinase inhibitor with a variety of effects on signal transduction pathways. Authors should be more cautious to attribute its effects to a single mechanism. Actually, in Figure 1 the effects on cell growth seem much larger than the increase in cell death detected and several, distinct death pattern are detected.
b) Moreover, in a clinical context, it is still matter of debate if the (limited) effects of sorafenib in HCC are due to direct effects on cancer cells or, rather, on the vascular network. This issue should be properly recognized in discussion.
c) Moreover, if sorafenib actually compromises tumor perfusion in vivo this should increase hypoxic areas and, according to the convincing evidence presented by authors, paradoxically limit its own capacity to elicit necrptosis in tumor cells. This intriguing hypothesis should be discussed.
d) The acronym AAG should be spelled out when cited for the first time.
e) RIPK1 seems much less affected than RIPK3 by the experimental treatments. Is this expected? Anyway, it should be properly reported in results description and adequately discussed.
f) Statistical information is quite superficial and only given in generic terms in the Methods section. It should be detailed for the single experiments in the legends to figures.
g) line 189. The intervention of intracellular proteases other than caspases (e.g. calpains) should not be excluded as major pathways for protein degradation.
h) Figure 7. It is not clear which changes are significant and which simply highlight a trend. Moreover, it is apparent that in Kaplan graphs most of he change is at later times of clinical course. Please discuss.
i) A discussion on what genetic changes could underlie the different pattern of gene expression assessed in Fig. 7 would be valuable.
k) I recommend a thorough and attentive language revision.
Author Response
Response to Reviewer 3 Comments
Point 1: Sorafenib is a multikinase inhibitor with a variety of effects on signal transduction pathways. Authors should be more cautious to attribute its effects to a single mechanism. Actually, in Figure 1 the effects on cell growth seem much larger than the increase in cell death detected and several, distinct death pattern are detected.
Response 1: Thank you for your comments. Sorafenib, as a multi-kinase inhibitor, has a significant effect on inhibiting the proliferation of HCC and promoting cell death. Many studies have reported the mechanisms of proliferation-related pathways of Sorafenib resistance. The specific mechanisms of hepatoma cell escaping from Sorafenib are not clear and need to be further studied. In our research, we found that HCC was resistant to Sorafenib under hypoxia via necroptosis blocking partly. The aim of this study is to investigate the mechanisms of Sorafenib resistance in necroptosis pathway under hypoxia.
Point 2: Moreover, in a clinical context, it is still matter of debate if the (limited) effects of sorafenib in HCC are due to direct effects on cancer cells or, rather, on the vascular network. This issue should be properly recognized in discussion.
Response 2: Thank you for this question. In fact, both the direct inhibitory effect of Sorafenib on tumor cells and on angiogenesis are very important. Many articles summarized mechanisms of Sorafenib. Sorafenib suppresses tumor angiogenesis and proliferation, and induces tumor cell apoptosis. References as follows. We have made proper statement in line 309-312 as you suggested.
Avila, M.A.; Berasain, C. Making sorafenib irresistible: In vivo screening for mechanisms of therapy resistance in hepatocellular carcinoma hits on mapk14. Hepatology 2015, 61, 1755-1757.
Jin, L.H.; Tabe, Y.; Zhou, Y.X.; Miida, T.; Andreeff, M.; Konopleva, M. Efficacy and mechanisms of apoptosis induction by simultaneous inhibition of pi3k with gdc-0941 and blockade of bcl-2 (abt-737) or flt3 (sorafenib) in aml cells in the hypoxic bone marrow microenvironment. Blood 2010, 116, 341-341.
Newell, P.; Toffanin, S.; Villanueva, A.; Chiang, D.Y.; Cabellos, L.; Lim, K.H.; Lesmes, P.M.; Yea, S.; Peix, J.; Deniz, K., et al. Mechanisms of ras pathway activation in hcc patients, and combination therapies blocking ras (sorafenib) and mtor pathways (rapamycin) in vivo. Hepatology 2008, 48, 983a-983a.
Thabut, D.; Routray, C.; Lomberk, G.; Shergill, U.; Glaser, K.; Huebert, R.; Patel, L.; Masyuk, T.; Blechacz, B.; Vercnocke, A., et al. Complementary vascular and matrix regulatory pathways underlie the beneficial mechanism of action of sorafenib in liver fibrosis. Hepatology 2011, 54, 573-585.
Point 3: Moreover, if sorafenib actually compromises tumor perfusion in vivo this should increase hypoxic areas and, according to the convincing evidence presented by authors, paradoxically limit its own capacity to elicit necrptosis in tumor cells. This intriguing hypothesis should be discussed.
Response 3: Thank you for your comments. The emergence of Sorafenib resistance is a gradual process. After a long-term use of Sorafenib, ischemic and hypoxic will appeared in tumor area due to the inhibition of microvessels. The inhibition of Sorafenib on tumor cells is weakened, then drug resistance is further developed. In our study, we found that MLKL was degraded under hypoxia. Necroptosis was blocked, which result in Sorafenib resistance. We have added some explanation in discussion of line 312-315 as you advice.
Point 4: The acronym AAG should be spelled out when cited for the first time.
Response 4: Thanks for your reminder. The full name of demethoxygeldanamycin (17-AAG) is formally appeared in line 20 for the first time.
Point 5: RIPK1 seems much less affected than RIPK3 by the experimental treatments. Is this expected? Anyway, it should be properly reported in results description and adequately discussed.
Response 5: Thank you for your comments. Both RIPK1 and RIPK3 are key roles in necroptosis signaling pathway. In this research, when Sorafenib-induced necroptosis occurs, both RIPK1 and RIPK3 were activated. As shown in Fig 1E, Fig 1G, and Fig 3A, the expression of RIPK1 and RIPK3 were both increased obviously after Sorafenib treated. Statistical analysis did not show significant difference between RIPK1 and RIPK3. In further study we will test the mRNA levels of RIPK1 and RIPK3 for accurate analysis.
Point 6: Statistical information is quite superficial and only given in generic terms in the Methods section. It should be detailed for the single experiments in the legends to figures.
Response 6: Thanks for your kind comments. We have detailed for the single experiments in the legends of figures.
Point 7: line 189. The intervention of intracellular proteases other than caspases (e.g., calpains) should not be excluded as major pathways for protein degradation.
Response 7: Thank you for your comments. There are indeed some another degradation pathways except for proteasome pathway, caspase pathway, and autophagy-lysosome pathway. In general, most proteins are mainly degraded by above manners. References as follows. Therefore, proteasome inhibitor MG132 was used to detect proteasome degradation pathway. Caspase inhibitor z-VAD-fmk was applied to invistigate caspase degradation pathway. Chloroquine was chosen to reveal autophagy degradation pathway We concluded that autophagolysosomal pathway may be the main degradation manner of MLKL.
Kihara, A.; Akiyama, Y.; Ito, K. Different pathways for protein degradation by the ftsh/hflkc membrane-embedded protease complex: An implication from the interference by a mutant form of a new substrate protein, ycca. J Mol Biol 1998, 279, 175-188.
Acquaviva, C.; Bossis, G.; Ferrara, P.; Brockly, F.; Jariel-Encontre, I.; Piechaczyk, M. Multiple degradation pathways for fos family proteins. Ann Ny Acad Sci 2002, 973, 426-434.
Carvalho, P.; Goder, V.; Rapoport, T.A. Distinct ubiquitin-ligase complexes define convergent pathways for the degradation of er proteins. Cell 2006, 126, 361-373.
Dikic, I. Proteasomal and autophagic degradation systems. Annual review of biochemistry 2017, 86, 193-224.
Hadji, A.; Clybouw, C.; Auffredou, M.T.; Alexia, C.; Poalas, K.; Burlion, A.; Feraud, O.; Leca, G.; Vazquez, A. Caspase-3 triggers a tpck-sensitive protease pathway leading to degradation of the bh3-only protein puma. Apoptosis : an international journal on programmed cell death 2010, 15, 1529-1539.
Point 8: Figure 7. It is not clear which changes are significant and which simply highlight a trend. Moreover, it is apparent that in Kaplan graphs most of the change is at later times of clinical course. Please discuss.
Response 8: Thank you for your comments. The results in Figure 7A presented the relationship of overall survival. The most significant change appeared in HSP90α, the p value is 0.0027. In Figure 7C-E, the differences in normal liver tissues and cancer tissues at later times of clinical course were showed. The figures from datebase that includes all the liver cancer stages. And we also observed the change of HSP90α, HIF1α, RIPK1, and RIPK3 in differtent liver cancer stages.
HSP90α, HIF1α, RIPK1 expressions are positively associated with disease stages and RIPK3 is negative, the slight trend barely be seen.
Point 9: A discussion on what genetic changes could underlie the different pattern of gene expression assessed in Fig. 7 would be valuable.
Response 9: Thank you for this question. As a molecular chaperone, HSP90α play important roles in proliferation, apoptosis, and drug resistance of various biological behavior. It is a regulator of P53, HIF1α, P23, HSF and so on that can perform their functions in develpment of cancer. HSP90α regulate the HIF signaling pathway in some reports, references as follows. In our study, we also found that HSP90α play crucial role in necrptosis signaling pathway, After we reducing the expression of HSP90α via shRNA, the necrptosis pathway was recovered. Therefore, HSP90α could underlie the different pattern of other genes expression. We have added this explanation in discussion of line 299-305 as you advice.
Lei, J.; Chen, J.; Zhu, Z.; Li, B.; Xue, K.; Zhang, C.; Wang, G. Melanin concentrating hormone modulate glycolytic reprogramming for regulating nlrp3 inflammasome activation via hsp90-hif1-alpha in psoriasis. J Invest Dermatol 2018, 138, S164-S164.
Zhao, S.; Dicker, D.T.; El-Deiry, W.S. Dual inhibition of cdk1 and hsp90 destabilizes hif1 alpha and synergistically induces cancer cell death. Cancer research 2017, 77.
Point 10: I recommend a thorough and attentive language revision.
Response 10: Thanks for your kind advice. We have polished the language of our manuscript carefully.
Thanks again for all your comments and valuable suggestions of our manuscript.

Round 2
Reviewer 2 Report
Authors did not perform any new experiment to address my issue. The answers that authors wrote regarding my criticisms were not supported by providing any reference and the list of refences, in fact, is almost the same as the previous version of the mansucript.
Author Response
Dear Editors: Thank you very much for your letter and advice on our paper entitled "HSP90α mediates Sorafenib resistance in human hepatocellular carcinoma by necroptosis inhibition under hypoxia" (Manuscript ID: cancers-1031282). First of all, we appreciated the reviewers’ advice to our work and acceptance for the reply. Based on the suggestions, we have carefully revised the manuscript according to the reviewer#2’s comments and make full responds point to point. Fig. 5A-B with new data has been reconstructed in the revised manuscript. Our revisions are clearly highlighted, we using the "Track Changes" function in Microsoft Word and PDF, so that changes are easily visible to you and reviewer#2. The revised manuscript has been submitted. Please feel free to contact me if you have any questions. We hope everything goes well with you during this unprecedented time of difficulty while we all fight against the pandemic of COVID-19 and happy new year. Best Regards. Yours Sincerely, Li Zhao, School of Basic Medicine and Clinical Pharmacology, China Pharmaceutical University, Nanjing 211100, China. E-mail address: zhaoli@cpu.edu.cn

Round 3
Reviewer 2 Report
Agrree to review